# Tripartite organization of brain state dynamics underlying spoken narrative comprehension

**Lanfang Liu[1,2], Jiahao Jiang[3], Hehui Li[4], Guosheng Ding[3]***

[1]Department of Psychology, School of Arts and Sciences, Beijing Normal University at Zhuhai, Zhuhai, China; [2]Faculty of Psychology, Beijing Normal University, Beijing, China; [3]State Key Laboratory of Cognitive Neuroscience and Learning, Beijing Normal University & IDG/McGovern Institute for Brain Research, Beijing, China; [4]Center for Brain Disorders and Cognitive Sciences, School of Psychology, Shenzhen University, Shenzhen, China

## eLife Assessment

This study provides **important** insights into the brain activity and connectivity underlying speech comprehension, revealing three brain states. The authors present **compelling** evidence by leveraging hidden Markov modeling of fMRI data to link brain state dynamics to comprehension scores, though the functional role of these states remains under-explored. These findings advance our understanding of how brain state transitions in narrative comprehension relate to stimulus-specific features.

**\*For correspondence:**
dinggsh@bnu.edu.cn

**Abstract** Speech comprehension involves the dynamic interplay of multiple cognitive processes, from basic sound perception, to linguistic encoding, and finally to complex semantic-conceptual interpretations. How the brain handles the diverse streams of information processing remains poorly understood. Applying Hidden Markov Modeling to fMRI data obtained during spoken narrative comprehension, we reveal that the whole brain networks predominantly oscillate within a tripartite latent state space. These states are, respectively, characterized by high activities in the sensory-motor (State #1), bilateral temporal (State #2), and default mode networks (DMN; State #3) regions, with State #2 acting as a transitional hub. The three states are selectively modulated by the acoustic, word-level semantic, and clause-level semantic properties of the narrative. Moreover, the alignment with both the best performer and the group-mean in brain state expression can predict participants' narrative comprehension scores measured from the post-scan recall. These results are reproducible with different brain network atlas and generalizable to two datasets consisting of young and older adults. Our study suggests that the brain underlies narrative comprehension by switching through a tripartite state space, with each state probably dedicated to a specific component of language faculty, and effective narrative comprehension relies on engaging those states in a timely manner.

## Introduction

When listening to a speech, one adaptively samples information from external sound streams, converting them to linguistic expressions stored in the mental lexicon, and integrating those mental expressions with the internalized 'mental world' to infer the semantic-pragmatic interpretations and intentions (*Berwick et al., 2013*). Crucially, those cognitive processes do not occur one after another in a fixed sequence, but are interwoven and occur in a fluid, dynamic manner. At one moment, you

might detect the auditory cues such as volume and pitch in the speech. At another, you might recall memories or knowledge in relation to certain words just heard. To effectively understand the speech, you must flexibly and adaptively switch among those cognitive processes. The neural mechanism behind it is still elusive.

An emerging view suggests that flexible and adaptive cognitive functions arise from the dynamic brain, which transiently activates and coordinates distributed neural circuits in response to the changes in the external environment and internal demands (*Honey et al., 2018*; *Kelso, 2012*). To capture the complex neural dynamics occurring across large-scale systems of the brain, researchers have conceptualized the brain's activity as operating on a low-dimensional neural manifold. The dynamics of brain activity can then be modeled as a temporal trajectory within a latent state space, with each latent state characterized by a distinct pattern of brain activities and network connectivities (*Langdon et al., 2023*). Employing statistical techniques for modeling dynamic systems such as Hidden Markov Modeling (HMM), recent studies have begun to explore the brain dynamics involved in narrative comprehension (*Baldassano et al., 2017*; *Song et al., 2021*; *Tang et al., 2023*) or movie viewing (*van der Meer et al., 2020*; *Song et al., 2023*). It has been found that the whole brain systematically switches among a limited number of temporal clusters or latent states with distinct spatial features. Moreover, the switching of brain states was modulated by the time-varying stimuli features including event boundary (*Baldassano et al., 2017*) and movie annotations (*van der Meer et al., 2020*), and subjective experience including engagement (*van der Meer et al., 2020*), attention fluctuations (*Song et al., 2023*), emotional changes (*Tan et al., 2022*), and narrative integration (*Song et al., 2021*). Those findings demonstrate the functional relevance of brain state dynamics. Nevertheless, how neural state dynamics contribute to the different streams of cognitive processing that ebb and flow with the unfolding of speech is still elusive.

In this study, we explored how language comprehension arises from the dynamic interplay of large-scale brain networks. According to the psycholinguistic theory, the basic design of language faculty mainly comprises three modules (components): an external sensory-motor module, an internal conceptual-intentional module, and a basic linguistic module which represents mental expressions formed by syntactic rules and connects the other two modules (*Berwick et al., 2013*). Built upon this theory, we hypothesize that brain dynamics underlying narrative comprehensions would predominantly oscillate within a tripartite latent state space, with each latent state primarily dedicated to a specific component of language faculty. Furthermore, we hypothesize that effective speech comprehension would rely on engaging these states in a timely manner.

To test the above tripartite-state-space hypothesis, we collected fMRI data from 64 young adults as they listened to 10 min real-life narratives. The HMM was applied to model the dynamics of whole-brain network activities. We expect the dynamics of whole-brain network activities would be optimally characterized by three latent states with distinct activity patterns. Specifically, one state would mainly activate the auditory and sensory-motor areas, contributing to the perceptual analyses of external sound streams. The second state would mainly activate the language network and frontal-parietal network, contributing to linguistic encoding and information integration. The third state would mainly activate the default mode networks (DMN), contributing to internalized semantic-conceptual processing. Moreover, according to the proposed architecture of language components, we expect both the externally-oriented and internally-oriented states would be more likely to transit to the second state than directly transiting between each other. To further validate the functional nature of the three states, we investigated how the dynamic changes of state expression probabilities would be modulated by the temporal variation of speech properties. Three speech properties were analyzed, each reflecting progressively longer timescales of linguistic information: sound envelope, word-level semantic coherence, and clause-level semantic coherence. Based on previous findings that the timescales of information accumulation vary hierarchically from early sensory areas to higher-order areas (*Hasson et al., 2015*; *Lerner et al., 2011*), we expect the occurrence probabilities of three brain states would be selectively modulated by these speech properties corresponding to distinct timescales. Finally, to probe the behavioral significance of the timing of brain states, we examined whether the alignment of a participant with the best performer in the time courses of brain state expression could predict the participant's narrative comprehension score. To validate the robustness of the results, we also conducted all the analyses using a replication dataset consisting of older adults.

## Results

### The brain reliably and robustly switches through three latent states

We applied the HMM to infer hidden brain states in 64 participants as they listened to one of three 10 min narratives. The observed variables were BOLD signal time series of nine networks (*Figure 1—figure supplement 1*) obtained employing a state-of-the-art technique for cortical network communities detection (*Ji et al., 2019*). Two criteria were comprehensively considered to determine the optimal number of latent states for the HMM. The first was the effectiveness of a model in capturing and separating patterns in the data, which was assessed by the clustering performance of the model. The second was the degree to which it aligned with prior knowledge about the data, which was evaluated by the model's ability to classify the three narratives. These dual criteria ensure that the selected model would be both statistically robust and cognitively sensible (*Pohle et al., 2017*).

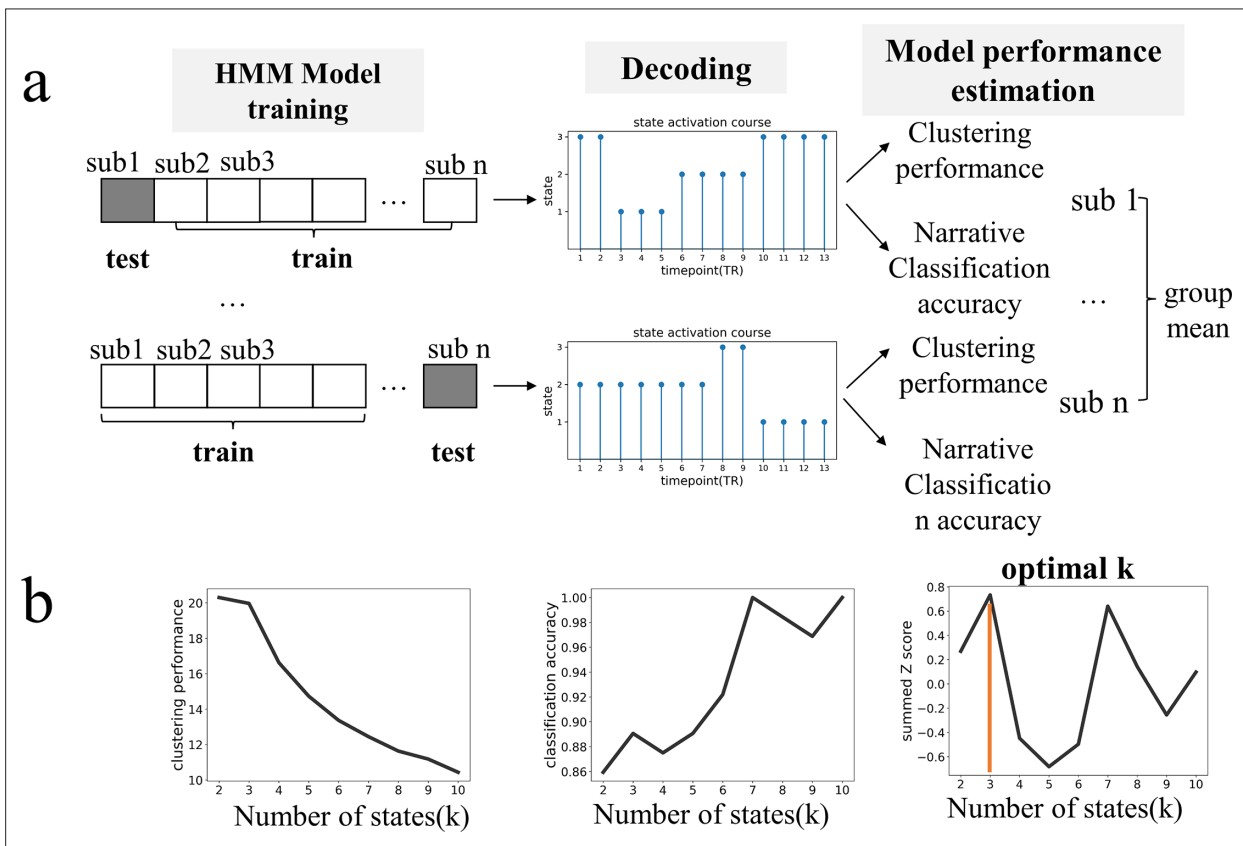

**Figure 1.** Identifying the optimal number of brain latent states based on the criterion of statistical robustness and cognitive sensibility. (**a**) For each candidate K ranging from 2 to 10, we trained a Hidden Markov Modeling (HMM) model on n-1 subjects and applied it to decode the time course of state expression for the test subject. The decoded time course was then used to compute a Calinski-Harabasz score, with a larger value indicating better clustering performance, and to decipher which narrative (out of three) was heard by the subject. The two measurements were first assessed at the individual level and then averaged across participants. (**b**) Model performance as a function of K. With the increase of K, the model's clustering performance tended to decline while the ability to decipher narrative contents tended to improve. We combined the two indices by converting them independently to *z* scores and summed them up. Notably, at K = 3, the summed *z*-score reached its highest point, therefore, it was set as the optimal number of latent states. See *Figure 1—figure supplement 2* for the replication of optimal K identification using a different whole-brain parcellation scheme and on a dataset of older adults.

The online version of this article includes the following figure supplement(s) for figure 1:

**Figure supplement 1.** Spatial maps of the nine networks derived from whole-brain parcellation using data from 64 participants engaged in narrative comprehension.

**Figure supplement 2.** Replicating the identification of the optimal number of states using a different scheme of brain network parcellation and on a dataset of older adults.

**Figure supplement 3.** Reconstructing the tripartite-state space from smaller states.

Across a range of candidate models with K from 2 to 10, the model's clustering performance tended to decrease with larger K, whereas the accuracy in classifying narrative contents tended to increase. The model with K = 3 achieved the best overall performance, quantified by summed *z* scores (*Figure 1*). When applying a different whole-brain parcellation scheme (Yeo-7 Networks atlas) to extract the brain time series used for HMM inference, we also found the model with K = 3 to be the optimal (*Figure 1—figure supplement 2*). Moreover, when examining the replication dataset wherein participants' age, narrative contents, as well as scanning duration differed substantially from those of the main dataset, we again found the model with K = 3 to be optimal (*Figure 1—figure supplement 2*). The robustness of findings suggests the tripartite state space likely captured some fundamental processes of the brain involved in narrative comprehension.

To further establish that the above tripartite-state organization was not trivial, we examined whether the three states could be reconstructed from smaller, more transient states. To this end, we applied a hierarchical clustering algorithm to the transition probability matrix derived from HMM models with 4, 10, and 12 states, and obtained three clusters for each. Those model orders were examined since they have been reported to be the optimal number in previous studies (*van der Meer et al., 2020*; *Song et al., 2021*; *Song et al., 2023*; *Vidaurre et al., 2017*). In this approach, states assigned to the same cluster were more likely to switch within themselves than switching to states belonging to other clusters, and such clusters have been called metastates in the literature (*Vidaurre et al., 2017*). We then examined whether there was significant and exclusive correspondence between the clustered states and the three target states (from the HMM with K = 3). There was a clear and exclusive correspondence between clusters reconstructed from both the 4-state and 10-state models and the predefined target states in terms of activity patterns ($r_{(6)}$ ranges from 0.72 to 0.98) (*Figure 1—figure supplement 3*). The timing of state expression was also well aligned between the reconstructed model and the target model (more than 77% overlap across a total of 19, 200 time points for 64 participants). For the 12-state model, we also found two reconstructed clusters resembling State #1 and State #3. Probe on the cluster that deviated most from the target states showed that it consisted of nine states, possibly capturing too many nuanced patterns of neural dynamics. Indeed, when splitting this 'big' cluster into two smaller ones, one of them demonstrated significant similarity to State#2 ($r_{(7)}$=0.75) . These findings suggest the tripartite-state organization is not trivial, but likely reflect some fundamental processes of brain dynamics.

## Three latent brain states have distinct spatial features

For each state, the HMM estimated its activity loadings on the nine networks and a functional connectivity matrix between these networks. We found the three latent states exhibited distinct activity patterns corresponding to the neural substrates for the three language components as suggested by the theory (*Berwick et al., 2013*). The first state (State #1) was characterized by relatively high activities in the auditory and somatomotor networks, along with low activities in the DMN and the cognitive control network (*Figure 2*). This state seems to be associated with the external sensory-motor module of language faculty. The second state (State #2) was characterized by relatively high activities in the language and the frontal-parietal networks whereas low activities in the somatomotor and auditory networks, seemingly being associated with the basic linguistic component. The third state (State #3) was characterized by relatively high activities in the DMN and frontal-parietal networks whereas low activities in the auditory and language networks, seemingly being associated with the internal conceptual-intentional component. We observed similar activity patterns of latent states when using the Yeo-7 network atlas for brain parcellation (*Figure 2—figure supplement 1*), and in the replication dataset consisting of older adults (*Figure 2—figure supplement 2*).

## State #2 acts as a 'transitional hub' with high functional integration

According to the linguistic theory, the module for linguistic representation is located in the middle of the external and internal modules, having direct interactions with the other two modules (*Berwick et al., 2013*). If the hypothesis that the three brain states were associated with each module of the language faculty holds, we expect that State #1 and State #3 would be more likely to switch to State #2 than switching directly to each other. Moreover, the brain occupied by State #2 would exhibit the highest degree of information integration.

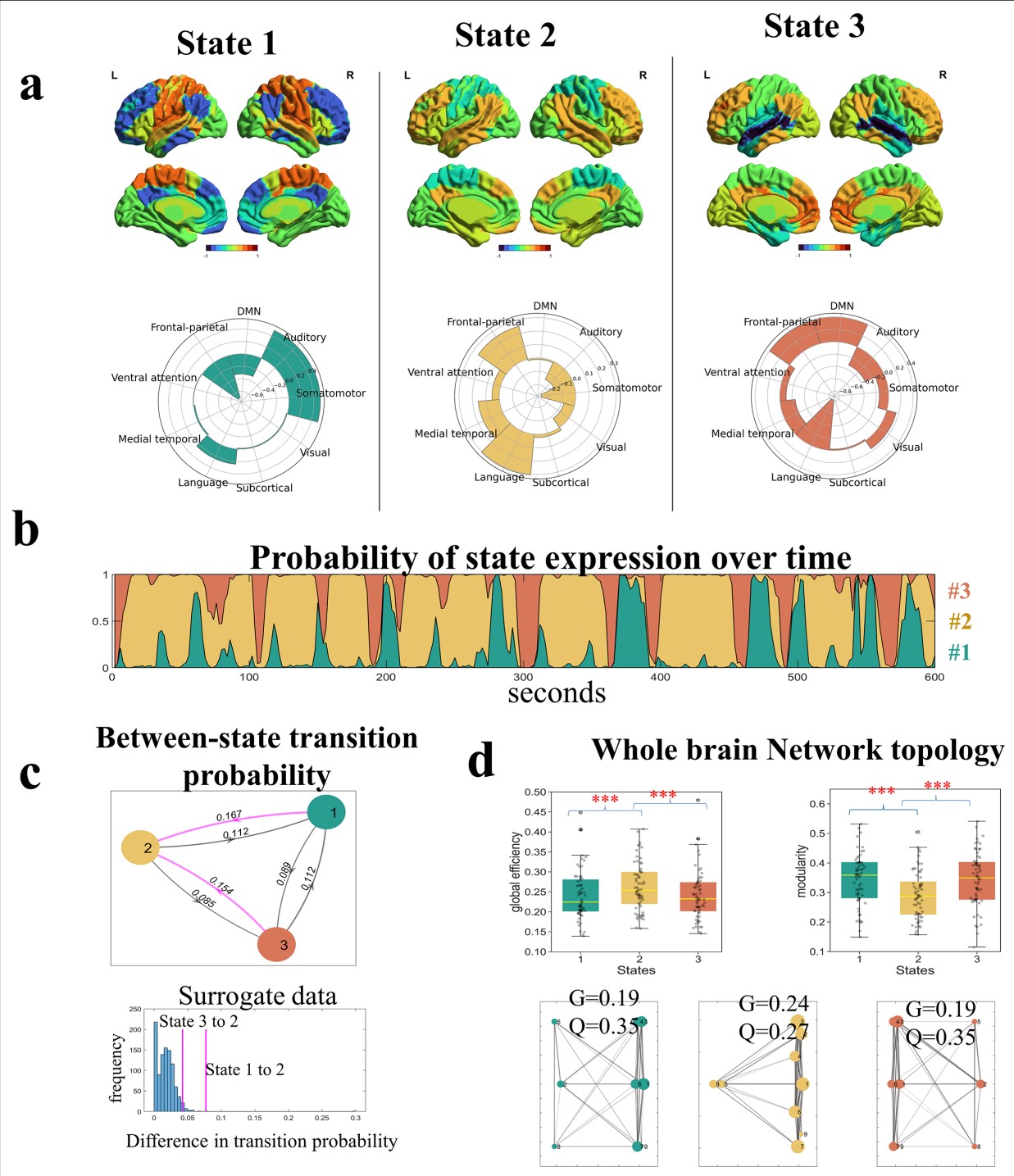

**Figure 2.** Spatial and temporal features of latent states revealed by Hidden Markov Modeling (HMM). (**a**) The activity loadings of each state on the nine networks. These activity values preserved the scale of the original BOLD time courses which were z-score normalized. (**b**) The ebb and flow of state expression over the time course of narrative comprehension, plotted using data from a representative participant. The curves of the three states are stacked showing the relative strength of activation probability at each time interval. (**c**) Between-state transition probabilities. Both State #1 and State #3 were more likely to switch to State #2 (denoted by the pink lines) than switching directly to each other. The differences in transition probabilities were larger than most of the instances from surrogate data. (**d**). Topological properties of whole-brain networks when occupied by each of the three states. Brain occupied by State #2 demonstrated the highest global efficiency (**G**) and the lowest modularity (**Q**). The upper panel shows the results of graph constructed using state-specific time series extracted from individual participants (N=64). The lower panel shows results of graph constructed using functional connectivity (FC) matrix derived from the HMM. The error bars indicate standard error of the mean. *$p < 0.05$, **$p < 0.01$, ***$p < 0.005$ for $t$-tests. See *Figure 2—figure supplements 1 and 2* for the replication with a different brain parcellation scheme and with the dataset of older adults.

*Figure 2 continued on next page*

*Figure 2 continued*

The online version of this article includes the following figure supplement(s) for figure 2:

**Figure supplement 1.** Replicating the spatial and temporal features of tripartite states using Yeo's 7-network atlas for brain network parcellation.

**Figure supplement 2.** Replicating the spatial and temporal features of tripartite states on the dataset of older adults.

To test the first prediction, we examined the between-state switching matrix inferred by the HMM, which showed the probabilities of a state at each timepoint transitioning to another or staying in the same state at the next timepoint. Consistent with our prediction, both State #1 and #3 were more likely to switch to State #2 than switching directly to each other, i.e., State #2 acted as a transitional hub. To confirm that this state-switching tendency was driven by meaningful processes rather than occurring by chance, we made surrogate data by having the nine-network time series circular shifted independently 1000 times. In each iteration, we carried out an HMM analysis with K = 3, and extracted the difference in transition probability if the inferred states exhibited a similar switching pattern as those from the experiment data. The results showed the differences in transition probabilities observed in our experiment, computed as $P_{(State\#1 \rightarrow State\#2)} - P_{(State\#1 \rightarrow State\#3)}$ and $P_{(State\#3 \rightarrow State\#2)} - P_{(State\#3 \rightarrow State\#1)}$, were respectively larger than 99.9% and 94.4% of instances from the surrogate data (*Figure 2*). In agreement with the between-state switching pattern, the brain spent most of the time on State #2 (mean FO = 46.7%), next on State #1 (mean FO = 29%) and the least on State #3 (mean FO = 24.3 %). The same pattern was found in the dwelling time, with a group mean of 15.29 s for State #2, 9.99 s for State #1, and 9.68 s for State #3. Both the FO and dwell time of State #2 were higher than 99.88% of instances in the surrogate data.

To test the second prediction, we applied the graph theoretical analyses to assess the global efficiency and modularity of the whole-brain networks when occupied by each of the three states. Consistent with our prediction, when occupied by State #2, the brain exhibited significantly higher global efficiency than when occupied by the other two states (*t* values >4.67, *ps* <10⁻⁴). An opposite pattern was found in network modularity (*t* values <–5.82, *ps* <10⁻⁶). These results indicate that, when occupied by State #2, the whole-brain networks were well connected to enable efficient information integration across distinct functional systems. In contrast, when occupied by State #1 and State #3, the whole-brain networks were well separated which enabled functional specialization. The findings were consistent whether using state-specific time series from individual participants to construct the FC matrix or taking the FC matrix derived from the HMM (*Figure 2*).

The between-state switching and topological properties are replicable using the different brain network parcellation schemes (*Figure 2—figure supplement 1*) and generalizable to the replication dataset consisting of the older adults (*Figure 2—figure supplement 2*).

## Expression of brain states is selectively modulated by narrative properties

To more directly establish the association of the three brain states to the theoretical language modules, we investigated how the expression of brain states would be modulated by changes in the stimulus properties as the narrative progressed. Three distinct stimuli properties presumably reflecting an increasingly deeper level of information conveyed by the narrative were extracted, including speech envelope, word-level semantic coherence, and clause-level semantic coherence.

Speech envelope captures the smoothed, slow amplitude fluctuations of the speech signal over time, which is the perceptual property of the stimuli. We observed a consistent positive correlation between the speech envelope and the expression probability of State #1 across participants, with a group mean of 0.035. This correlation was significantly greater than zero ($t_{(63)}$ = 2.67, *p* = 0.009, FDR corrected) and surpassed 99.8% of the 5,000 instances from permutation tests where the state time courses were randomly circularly shifted, yielding a mean of −0.0005 (*Figure 3*). A similar effect was observed for State #2, with a group mean *r*=0.034 ($t_{(63)}$ = 2.61, *p* = 0.011, FDR corrected).

Word-level semantic coherence was evaluated using cosine similarity between embedding vectors for each word and the preceding word. Among the three states, only the expression probability of State #2 showed a consistent correlation with word-level semantic coherence across participants, with a group mean of 0.030 (one-sample t-test against zero: $t_{(63)}$ = 2.48, *p* = 0.015, FDR corrected). This

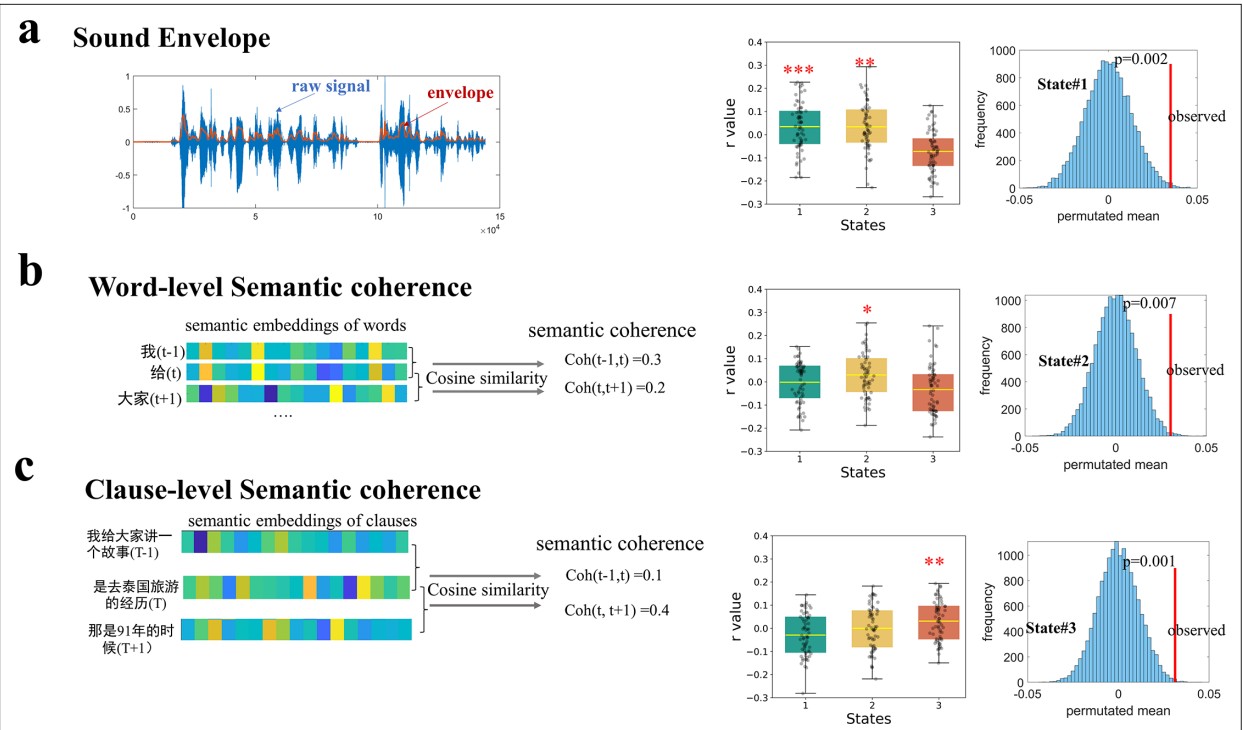

**Figure 3.** Selective modulation of state expression by different narrative features. (**a**) The expression probability of State #1, as well as of State #2, was positively modulated by the temporal envelope of speech. The blue shaded area represents the original amplitude of a speech signal clip, while the orange line shows the smoothed, low-frequency contour of amplitude changes over time (i.e. the sound envelope) (**b**) Only the expression probability of State #2 was modulated by word-level semantic coherence. (**c**) Only the expression probability of State #3 was modulated by clause-level semantic coherence. Semantic coherence was measured by cosine similarity between the embeddings (obtained by BERT) of each word (or clause) and the word (clause) immediately before it. Those effects were greater than most instances from permutation where the time courses of state expression were randomly shuffled in a circular manner 5000 times. The error bars indicate standard error of the mean. *$p < 0.05$, **$p < 0.01$, ***$p < 0.005$ for two-tailed $t$-tests (N=64). See *Figure 3—figure supplement 1* for the replication of the modulation effects with a different brain parcellation scheme and with the dataset of older adults.

The online version of this article includes the following figure supplement(s) for figure 3:

**Figure supplement 1.** Replicating the modulation effects using a different scheme of brain network parcellation and on a dataset of older adults.

group's mean value was significantly greater than 99.28% of instances from permutation tests, which produced a mean $r = 0.001$.

Clause-level semantic coherence was evaluated using cosine similarity between embedding vectors for each clause and the preceding clause. Among the three states, only the expression probability of State #3 exhibited a consistent correlation with clause-level semantic coherence, with a group mean of 0.031 (one-sample t-test against zero: $t_{(63)} = 2.89$, $p = 0.005$, FDR corrected). This group's mean value was significantly greater than 99.8% of instances from the permuted data, which yielded a mean $r = 0.0001$.

We noted that the correlation value between individual brain state dynamics and speech features was relatively low (group averaged $r \approx 0.03$), which is also the case in previous studies (e.g. *Fernandino et al., 2022*; *Oota et al., 2022*). This low correlation is likely due to the high level of noise inherent in fMRI data. Brain activity fluctuations are shaped by a variety of factors, including task-related cognitive processing, internal thoughts, physiological states, as well as arousal and vigilance. Additionally, the speech features we measured may account for only a small portion of the cognitive processes underlying the task. As a result, the variance in speech features may only explain a limited portion of the overall variance in brain state fluctuations. Nonetheless, the brain-stimuli correlation achieves statistical significance, and is reproducible across different populations and narratives, providing robust support for the reliability of the results.

In sum, the selective modulation by the different aspects of narrative properties provides further evidence supporting the functional relevance of three latent brain states to different language

components. These results are replicated with the different brain network atlas (*Figure 3—figure supplement 1*). On the older-adult dataset, we also observed selective modulation effects of speech envelope on State #1 and word-level semantic coherence on State #2; however, no modulation effect of clause-level semantic coherence was found (*Figure 3—figure supplement 1*).

## Inter-subject correlation in brain state dynamics predicts task performance

The above results have demonstrated the functional relevance of the tripartite state space to narrative comprehension. Next, we tested the hypothesis that effective narrative comprehension would rely on engaging these states in a timely manner. To tackle this question, we measured the alignment of brain state fluctuation between each participant (except for the best performers) with that of the best performer(s). Then we used the inter-brain alignment index to relate to participants' comprehension scores, which were measured from the post-scan recall of the stories scored by two independent raters. The best performers were the ones (or those) who achieved the highest comprehension score within the subgroup of participants exposed to the same narrative. The rationale is that, if effective comprehension relies on the brain to turn into specific patterns at the right times, the best performer would demonstrate the most 'accurate' pattern. Consequently, participants whose brain state fluctuations deviated more (or less alignment) from the 'accurate' pattern were anticipated to perform less effectively in the task.

As anticipated, alignments with the best performers in both the State #1 and State #2 were significantly correlated with participants' comprehension scores (Pearson's $r_{(54)}$ = 0.31 and 0.37, respectively). A marginally significant correlation was also found in the alignment of State #3 ($r_{(54)}$ = 0.22, $p$ = 0.10). We also examined participants' alignment with the group-mean time courses of brain state

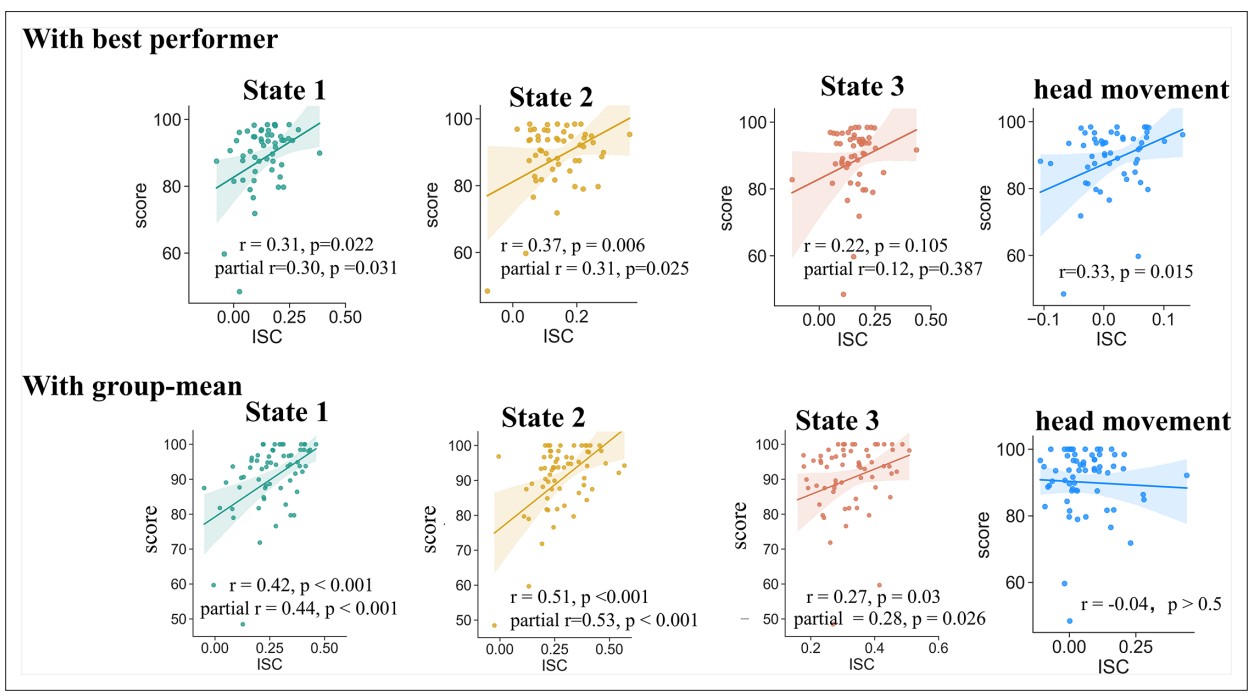

**Figure 4.** Correlation of state expression with behavior. Participants' alignment with both the best performer(s) and the group mean in terms of brain state expression predicted their narrative comprehension scores. The alignment with the best performer in head movement trajectory, which probably reflected inter-subject similarity in the fluctuation of task engagement or attention, also correlated with narrative comprehension. After adjusting this effect using partial correlation, the significant correlations between inter-subject alignment in states expression and narrative comprehension still existed. The results were replicated with a different brain parcellation scheme (*Figure 4—figure supplement 1*), but irreproducible in the older-adults dataset, probably due to substantial heterogeneity among them (*Figure 4—figure supplement 2*).

The online version of this article includes the following figure supplement(s) for figure 4:

**Figure supplement 1.** Replicating brain-behavior correlation using the Yeo's 7-network parcellation scheme.

**Figure supplement 2.** Comparison in inter-subject correlation (ISC) in brain state dynamics between the young and older age groups.

expression, which is supposed to represent the systematic response to the stimulus (*Nastase and Gazzola, 2019*). Even stronger correlations were found between individual-to-group alignments and comprehension scores in all three states ($r_{(62)}$ = 0.42, 0.51, 0.27 for State #1, #2, and #3, respectively; *Figure 4*).

In previous studies, the similarity of brain activities across subjects has usually been interpreted as reflecting the inter-subject similarity in the fluctuation of task engagement or attention (*Nanni-Zepeda et al., 2024*; *Ohad and Yeshurun, 2023*), which in turn may be associated with the individual similarity in task performance. We examined whether the above association of inter-subject alignment in brain states with behavior was merely an epiphenomenon of overall task engagement. It is well known that continuous self-reports on task engagement may severely disrupt the ongoing processing of prolonged naturalistic stimuli. As an alternative, studies have demonstrated that head movement serves as a reliable time-resolved indicator for task engagement, with greater task engagement accompanied by decreased movement (*Ballenghein et al., 2019*; *Greipl et al., 2021*; *Kaakinen et al., 2018*). Leveraging this, we computed inter-subject correlations (ISC) in the trajectory of head movement (quantified by framewise displacement) during the fMRI scanning as a proxy for inter-subject similarity in task engagement. Congruent with our assumption, similarities with the best performers in terms of head movement trajectory were indeed positively correlated with participants' comprehension scores ($r_{(54)}$ = 0.33, $p$ = 0.01; *Figure 4*). After adjusting the effect of head movement by applying partial correlation, the positive correlations between the inter-brain alignment with the best performers and comprehension scores remained significant for State #1 (partial $r_{(53)}$ = 0.30, $p$ = 0.03) and State #2 (partial $r_{(53)}$ = 0.31, $p$ = 0.025), except for State #3 (partial $r_{(53)}$ = 0.12, $p$ = 0.39). These findings suggest that the inter-subject alignments in brain states were unlikely merely the byproduct of shared levels of task engagement, but instead reflected the commonality in neural processes that directly influence narrative comprehension.

As a comparison, we investigated whether individual differences in the FO and dwell time of latent states were associated with individual differences in narrative comprehension. No significant result was found on any of the three states ($r$ values <0.15, $p$s >0.23). Taken together, these findings suggest that timely engagement with specific brain states, rather than the overall magnitude of engagement in those states, is crucial for narrative comprehension.

These findings were replicable with a different scheme for brain network parcellation (*Figure 4—figure supplement 1*). However, in the dataset of older adults, we did not find significant positive correlations between inter-subject alignment in brain states and narrative comprehension. This may be due to substantial heterogeneity among older adults. Despite engaging in the same task, older adults exhibit considerable inter-subject variation in cortical morphology, function, and brain metabolism (*Yu, 2024*). These factors can diminish the inter-subject correlation of brain state dynamics — indeed, ISCs among older adults were significantly lower than those among younger adults (*Figure 4—figure supplement 2*) — and potentially reduce ISC's sensitivity to individual differences in task performance.

## Comparison between conditions

The above results have revealed a tripartite latent space of whole-brain dynamics, with each state probably subserving a different cognitive component underlying narrative comprehension. Is this temporospatial organization a task-free, intrinsic organization of the dynamic brain, or mainly driven by language processing? To address this question, we compared the brain states involved in the narrative comprehension task with those of the same participants when they listened to an unintelligible narrative (told in Mongolian, MG) and during rest. Note, the involvement in linguistic computations decreased monotonically across the three conditions.

HMMs with K = 3 were conducted separately for the resting and MG conditions. To establish correspondence between states in the resting and MG conditions with those in the task condition, we examined the correlation in network activities for each state identified in the rest/MG (treated as 'candidate state') with each of the three states identified in the task (treated as 'predefined state'). The predefined state that a candidate state most closely resembled was designated as its matching state.

For the resting condition, two candidate states showed the highest similarity to State #2 in the task condition. The most similar candidate ($r_{(7)}$ = 0.778) was labeled as State #2. The other candidate was assigned to the predefined state (State #3) with the second-highest correlation, denoted with a prime symbol (State #3') to indicate this confusion. The last candidate state matched best with State

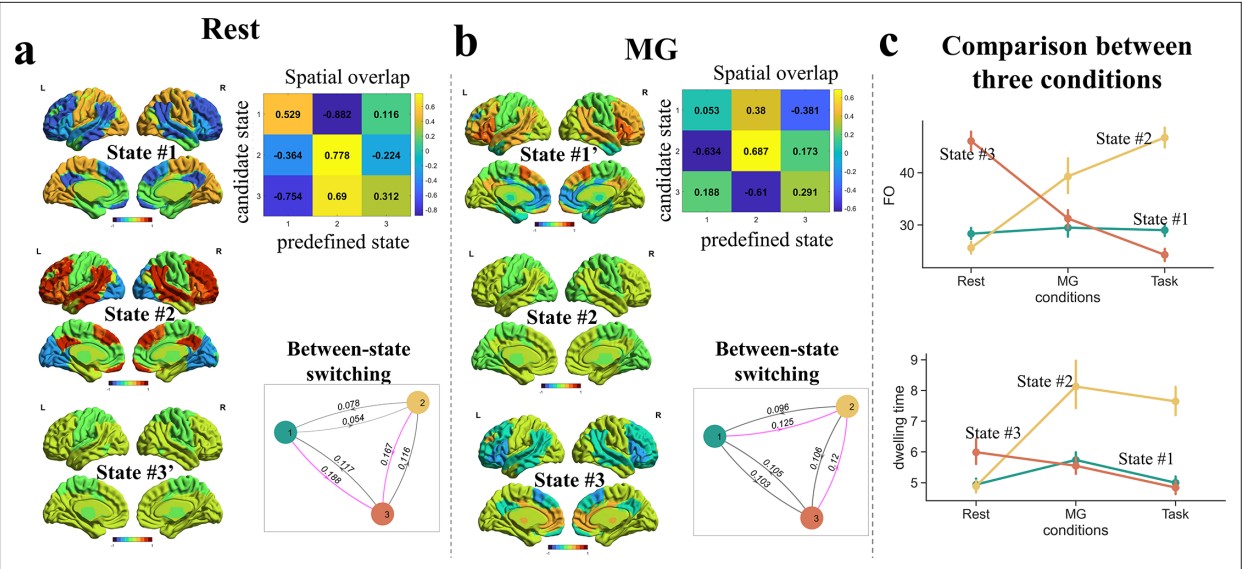

**Figure 5.** Comparisons of brain states across conditions. (**a**) During the rest period, the activity patterns of latent states were similar to those during narrative comprehension, but State #3 became the transitional hub. (**b**) When listening to the unintelligible narrative (in Mongolian, MG), the activity patterns of latent states varied substantially from that during narrative comprehension, but State #2 was still the transitional hub. (**c**) The fractional occupation of State#2 increased with greater involvement in linguistic computations, while that of State#3 decreased. A similar pattern was found on the dwelling time of states. The error bars indicate standard error of the mean.

#1 without any confusion. Analysis of the between-state transition matrix revealed that, unlike in the task condition, State #3 acted as the transitional hub in the resting condition (***Figure 5a***).

For the MG condition, two candidate states also showed the highest similarity to State #2. The most similar candidate ($r_{(7)} = 0.687$) was labeled as State #2, while the other was labeled as State #1'. The final candidate state best matched with State #3 without any ambiguity. Consistent with the task condition, State #2 served as the transitional hub under the MG condition (***Figure 5b***).

Notably, the FOs of State #2 monotonically increased across the three conditions: resting < MG (incomprehensible speech) < task (narrative comprehension). In contrast, the FOs of the State #3 monotonically decreased: resting > MG > task (***Figure 5c***). A similar pattern was found on the dwelling time. These findings provide additional evidence supporting that State #2 was associated with linguistic computations, while State #3 was associated with internalized mental activities. Together, these results suggest that the tripartite latent space of whole-brain dynamics is mainly driven by language processing.

## Discussion

Speech comprehension is a sophisticated cognitive task that requires the dynamic interplay of various processes, from basic sound perception to complex semantic-pragmatic interpretations, all of which are fluidly coordinated in real-time as the speech unfolds. Here, we explored how the brain transiently activates and coordinates distributed neural networks to support the diverse cognitive streams underlying spoken narrative comprehension. Applying HMM, we found that the brain reliably and robustly switched through three latent states, which were characterized respectively by high activities in the sensory-motor (State #1), bilateral temporal-frontal (State #2), and DMN (State #3) regions. Among them, State #2 occurred most frequently, acted as a 'transitional hub,' and was characterized by the highest level of functional integration. Furthermore, the three states were selectively modulated by the perceptual, word-level semantic, and clause-level semantic properties of the speech. Importantly, participants' alignments with both the best performers and the group-mean on the time courses of brain states expression predicted their narrative comprehension scores, indicating effective speech comprehension relies on engaging the specific brain states in a timely manner. Finally, by comparing the comprehension task with the resting and the unintelligible speech conditions, we demonstrated that the tripartite latent state space was mainly driven by language processing.

A set of results convergently suggests that the tripartite state space is not incidental, but likely reflects a fundamental principle governing the brain dynamics underlying narrative comprehension. First, among a range of candidate models with the number of states ranging from 2 to 10, the model with K = 3 performed the best in terms of separating patterns in the data and decoding narrative contents, being both statistically robust and cognitively sensible. Second, the tripartite latent state space was replicable with different network atlas and generalizable to different datasets. Especially, despite that the two datasets vary substantially in terms of participants (young versus older adults), the contents of narratives, and data length, the two groups still exhibited highly similar brain temporal organization that was best captured by the three latent states. Moreover, the spatial and temporal patterns of the tripartite state space can be hierarchically reconstructed from more nuanced state patterns, being 'metastates' of the brain.

Intriguingly, the characteristics of the three latent states align well with the theoretical framework concerning the basic design of language faculty (*Berwick et al., 2013*). The State #1, which was featured by relatively high activities in the auditory and somatomotor networks and more likely to occur when speech sounds were louder, probably corresponds to the external sensory-motor component of language faculty. State #2, which was featured by relatively high activities in the bilateral temporal and the frontal-parietal networks and more likely to occur when the inputting word was semantically related to the word immediately before it, probably corresponds to the basic linguistic component. State #3, which was featured by relatively high activities in the DMN and frontal-parietal networks and more likely to occur when the inputting clause was semantically related to the clause immediately before it, probably corresponds to the higher-level semantic-conceptual component. Moreover, State #2 acted as a transitional hub, with both State #1 and State #3 were more likely to switch to it than switching directly within them. This directed switching pattern was not incidental, as the observed discrepancy in switching probability was greater than in most instances from the surrogate data. Additionally, when occupied by State #2, whole brain networks exhibited a higher level of information integration (quantified by global efficiency) than when occupied by the other two states. These patterns are also consistent with the theoretical prediction that the basic linguistic module is located in the middle of the external and internal modules, having direct interactions with the other two. Collectively, these findings demonstrate the specific relationship between the tripartite brain latent states and three critical components of language cognition, going beyond the account of arousal or attentional fluctuation for brain state dynamics (*van der Meer et al., 2020*; *Song et al., 2023*; *Taghia et al., 2018*).

The activity patterns of the brain's latent states associated with each of the theoretical components of language faculty are consistent with the findings from earlier studies that have mainly focused on the averaged brain activities over time (*Ferstl et al., 2008*; *Price, 2012*). Extending prior work, our study may provide novel insights about how the different streams of cognitive processing are temporally organized with the unfolding of speech. Specifically, the time-varying probabilities of latent states indicate that the associated cognitive processes underlying speech comprehension may not operate in parallel with equal priority or occur one after another. Instead, while all processes are simultaneously engaged, one process dominates over others and this dominance changes over time, taking the form of mode-switching. This is consistent with the emerging view that internal/external switching processes of neural circuits drive learning (*Honey et al., 2018*). Furthermore, we found it was the alignment with the best performer (and the group-mean) in the time courses of state expression, rather than the overall occupancy of latent states, that positively correlated with participants' task performance, suggesting recruiting these states in a timely manner is the key to effective speech comprehension. These findings may provide a useful guide to understand the development of language ability as well as language disorders.

Our study provides a unifying perspective for two prevailing approaches aiming to understand how the brain produces cognition. The modular approach postulates the brain areas to act as independent processors for a specific aspect of complex cognitive functions, contributing to much of our current knowledge of the relationship between the brain and behavior. However, this approach has been criticized for ignoring the multifunctionality of brain structures (*Fuster, 2000*). Alternatively, the network approach, which has been growing rapidly in recent years, posits that cognitive functions arise from dynamic interactions within and between distributed brain systems (*Bressler and Menon, 2010*). While revealing valuable insights into the operation rules of the brain, the network approach seems to

only provide a general descriptive model. It lacks a mechanism that accounts for how the interactions of large-scale brain networks give rise to the different streams of information processing involved in a cognitive task. Here, by demonstrating the multistability of large-scale brain networks and establishing the close relationship between specific latent states to specific language components, our study raises a hypothesis that could reconcile the modular and the dynamic network approaches to understand the brain function. Specifically, for a given task, the brain follows modular organization where different regions specialize in specific functions. However, the importance of these regions dynamically changes in response to external environment and internal demands. Accordingly, goal-directed behaviors arise from the precise temporal coordination of different functional modules (*Vyas et al., 2020*). To test this possibility, future studies could combine fMRI and neuroregulation techniques and assess the change in state dynamics and behavioral performance as a result of an intervention.

## Conclusion

In sum, our study reveals that the brain involved in narrative comprehension predominantly oscillates within a tripartite latent state space. The spatial and topological characteristics of these states correspond well to the three core components of language faculty as specified in the theory (*Berwick et al., 2013*). Moreover, we demonstrate that effective speech comprehension relies on engaging these brain states in a timely manner. These results are largely reproducible with different brain network parcellation schemes, and generalizable to two replication datasets consisting of young and older adults. The findings establish the link of brain dynamics with both ongoing cognitive processing and behavioral outcomes, providing a mechanistic account of how language comprehension arises from the dynamic interplay of large-scale brain networks.

# Materials and methods
## Participants and experiment procedure

The main dataset consisted of 68 Chinese college students (33 males, aged 19–27 y) scanned with fMRI while listening to a 10 min narrative in Chinese. Data of four participants were omitted from further analyses due to excessive head movements (>3 mm or 3 degrees).The speech played to each participant was randomly chosen from three real-life stories told by a female college student. These stories described her experiences with a graduate admission interview, a hiking trip, and her first time taking a flight.

The replication dataset came from 30 healthy older adults (12 males, aged 53–75 y) recruited from the residential community near the college. During fMRI scanning, each participant listened to two real-life stories presented with and without background noise. Those stories were told by a 62-year-old woman in a Chinese dialect, describing her experience in Thailand and riding a warship, respectively. After omitting 10 runs with large head movements (>3 mm or 3 degrees), a total of 50 runs of fMRI scans were included for subsequence analyses.

For both datasets, the recordings were collected when the speakers were scanned with fMRI. To obtain audio recordings with good quality, we used a noise-canceling microphone (FOMRI-III, Opto-acoustics Ltd., Or-Yehuda, Israel) positioned above the mouth of the speaker and further de-noised the audio recordings offline using Adobe Audition 3.0 (Adobe Systems Inc, USA).

After the scanning session, participants were asked to retell the story in detail. Following the free recall, experimenters asked additional questions from a prepared list about details not previously mentioned. The list contained 9–10 questions for each story, which addressed key events from different parts of the narrative (e.g. 'Can you describe the preparations she made for the hike?' or 'What happened on her way to the hotel?'). Correct answers are needed to cover critical details, including actions, characters, locations, time, and motivations. Two independent raters listened to the recordings of participants' free recall and their responses to the experimenters, assigning a score for each question based on the completeness of the key points provided. The comprehension score for each participant was calculated as a percentage of the total possible points. Given the high inter-rater reliability (Spearman's $r_{(62)} = 0.74$), the average score from both raters was used to quantify each participant level of story comprehension. The audio files of those stories, along with their corresponding transcriptions and sample questions used in the comprehension test, are available at: https://github.com/liulanfang11/Tripartite-brain-states/upload/main.

This research was approved by the Reviewer Board of Beijing Normal University in China (ICBIR_A_0002_009). Written informed consent was obtained from all participants before the study. The dataset has been used in our previous work addressing different questions (*Liu et al., 2022*; *Liu et al., 2020*).

## MRI acquisition and preprocessing

We used a 3T Siemens Trio scanner in the MRI Center of the Southwest University of China to collect imaging data. Functional images were acquired employing a gradient echo-planar imaging sequence with the specified parameters: repetition time = 2000 ms, echo time = 30 ms, flip angle = 90°, field of view = 220 mm$^2$, matrix size = 64 × 64, 32 interleaved slice, voxel size = 3.44 × 3.44 × 3.99 mm$^3$. Structural images were acquired using a MPRAGE sequence with the following parameters: repetition time = 2530 ms, echo time = 3.39 ms, flip angle = 7°, FOV = 256 mm$^2$, scan order = interleaved, matrix size = 256 × 256, and voxel size = 1.0 × 1.0 × 1.33 mm$^3$. The preprocessing pipeline included slice-timing correction, spatial realignment, co-registration to the individual participants' anatomical maps, normalization to the Montreal Neurological Institute (MNI) space, resampling into a 3 × 3 × 3 mm$^3$ voxel size, and smoothing (FWHM = 7 mm). The resulting images underwent additional processing, including detrending, nuisance variable regression, and high-pass filtering (1/128 Hz). Data with head movement exceeding 3 degrees or 3 mm were omitted from further analyses.

## Data analyses

### Whole-brain parcellation

The inference for brain dynamic states was conducted at the whole-brain network level. Currently, most brain functional networks reported in the literature are made based on resting-state fMRI data. To better capture the brain network organization during the task, we conducted brain network parcellation applying a state-of-the-art method proposed by *Ji et al., 2019*, using data from the 64 participants engaged in narrative comprehension. This method employs multiple quality control metrics to ensure the stability and reliability of the network partition, and most importantly, uses parameter optimization guided by well-established neurobiological principles (e.g. the separation of sensory and motor systems).

First, the mean time series of 246 regions defined by the Brainnetome atlas (*Fan et al., 2016*) were extracted, and a functional connectivity (FC) matrix was computed for each participant and then averaged across them. This atlas covers both cortical and subcortical regions and is made based on both anatomical and FC patterns. Next, community detection was performed on the group-averaged FC matrix by applying the Louvain clustering algorithm in the Brain connectivity toolbox (https://sites.google.com/site/bctnet/). Three criteria were taken into account when determining the Gamma parameter in the algorithm, including (1) separation of primary sensory-motor network (visual, auditory and somatomotor) from all other networks (i.e. neurobiologically sensible); (2) high similarity of network partitions across nearby parameters (i.e. statistically stable); and (3) high within-network connectivity relative to between-network connectivity (i.e. high modularity).

A set of gamma values ranging from 1.2 to 2.5, with a step size of 0.01, were tested. For every tested gamma, we ran the algorithm 1000 times and measured how consistent a given partition was to every other partition using a z-rand score. Each z-rand score averaged across the iterations was then multiplied by its corresponding modularity score to find a modularity-weighted z-rand score. Finally, the gamma value of 2.5 was selected, as it corresponded to the peak of the modularity-weighted z-rand score and satisfied the three criteria, resulting in a plausible number of networks, including the primary sensory/motor networks. We implemented network detection using code published by a prior study (*Barnett et al., 2021*).

The network detection approach identified a total of 11 networks (*Figure 1—figure supplement 1*). Two networks were discarded due to comprising too few nodes (less than three) and thus nine networks were included for further analyses. By reference to the functional decoding results using NeuroSynth (https://www.neurosynth.org), we tentatively labeled the nine networks as the auditory, visual, somatomotor, bilateral language, medial temporal, frontal-parietal, ventral attention, subcortical, and default mode networks. To test the robustness of the findings, we also adopted the seven-network atlas (*Yeo et al., 2011*) for whole-brain parcellation and reconducted the main analyses.

## Brain state inference using Hidden Markov Model

We applied the HMM to infer latent brain states during narrative comprehension using the HMM-MAR toolbox (https://github.com/OHBA-analysis/HMM-MAR; *Vidaurre and Higgins, 2025*). The HMM assumes that the observed data is generated through a finite number of latent states, and each state can be characterized, respectively, by a multivariate Gaussian distribution with mean and covariance. The BOLD time series was first standardized within each participant and each network. Then HMM was fitted using concatenated data from all participants, such that unified brain states could be obtained.

The number of latent states (represented by 'K') is a crucial aspect of the HMM, and it needs to be predetermined before fitting the model. Two criteria were considered to determine the optimal K. The first was a model's clustering performance, which reflects how well the model can capture and separate different patterns in the data. The second criterion was how well the model aligned with existing knowledge about the data. This criterion was evaluated by the ability of a trained HMM model to decode the narrative content heard by unseen participants. This dual criterion ensures that the selected number of brain states (K) for the HMM is both statistically robust and cognitively meaningful (*Pohle et al., 2017*). The clustering performance and prediction accuracy were assessed through a leave-one-out cross-validation strategy. In this approach, we trained the HMM using data from all participants except one. For each candidate K, we repeated the training process 10 times, and the instance with the smallest free energy was selected for decoding the latent state sequence of the left-out participant (*Song et al., 2023*). Utilizing the decoded latent state sequence, along with the participant's network time series, we calculated the Calinski-Harabasz score as an indicator of the model's clustering performance, with a higher score indicating better clustering performance. Furthermore, to assess the model's decoding capability, we applied a K-nearest neighbor algorithm utilizing the decoded latent state sequences to classify which of the three narratives the left-out participant was listening to. A higher accuracy indicates the model has well captured the task information in the data. Both the Calinski-Harabasz score and narrative classification accuracy were acquired from each participant and then averaged across the group. To combine the two criteria, we first converted the Calinski-Harabasz score and narrative decoding accuracy independently to Z scores and then summed them up to create a single composite score.

We repeated the above cross-validation procedure across a range of K from 2 to 10. The K with the largest composite score was set to be the optimal number of HMM states representing the brain dynamics during the narrative comprehension task. Upon determining the optimal number of states, we reconducted the HMM on the data from all participants and chose the instance with the largest model evidence (lowest free energy) from 10 iterations as the final result.

## Reconstruction of latent states using hierarchical clustering

To demonstrate that those hidden states identified by the above analyses were not trivial but potentially reflected several fundamental processes of the dynamic brain, we explored whether those states can be reconstructed from smaller, more nuanced patterns using hierarchical clustering (*Vidaurre et al., 2017*). To this end, we applied an agglomerative hierarchical clustering algorithm to the transition probability matrix derived from HMM with K = 4, 10, and 12, respectively, and obtained three clusters from each. Next, we compared the clusters with the target states (i.e. those resulting from the HMM with K = 3) in terms of similarities in activity patterns (spatial overlap) and the time course of state expression (temporal overlap).

To evaluate the spatial overlap, we first averaged the activation values across those states belonging to the same cluster, merging them into a single new state. Then we assessed the similarity in the activity patterns between the merged states and the target states using Pearson's correlation. To evaluate temporal overlap, we first substituted states belonging to the same cluster with the newly formed one, then computed Jaccard Similarity between the sequences of the new states and the sequences of the targeted states.

## Analyses of brain state properties

The HMM model generated, for each state, a group-level activation map and a functional connectivity matrix, as well as a between-state transition probability matrix. With these parameters, the probability of each state being active (or expressed) at each time point and the most likely sequence of states were estimated for each participant using the Viterbi algorithm (*Rezek and Roberts, 2005*). Based on

the Viterbi path, the total time spent on each state over the entire duration (referred to as FO) and the duration for which a state continuously persisted before switching to another one (referred to as dwell time) were computed for each participant.

Next, we conducted a graph theoretical analysis to assess the degree of functional integration and segregation of the whole brain when occupied by a specific state. For each participant and each state, a weighted and undirected graph was constructed in which the nine networks were represented as nodes, and FCs estimated using network time series corresponding to the specific state were represented as edges. Employing the Brain Connectivity toolbox (*Whitfield-Gabrieli and Nieto-Castanon, 2012*), we computed network global efficiency as the measurement for functional integration. Functional segregation was measured by a network modularity score using the Louvain algorithm with a resolution parameter gamma = 1. Then t-tests were employed to examine the differences in these indices across the three states. For validation purposes, we additionally computed the two graph theoretical indices using the state-specific FC matrices estimated by the HMM.

## Surrogate data generation and permutation test

To ascertain that the trend in between-state transition was not by chance, we generated surrogate data by having the 9-network time series circular shifted independently. In this approach, the meaningful covariance between networks was disrupted while the temporal characteristics of the time series were retained (*Song et al., 2023*). On each permutated data, we conducted an HMM analysis with the optimal K (i.e. K = 3). If there were two states where both showed a higher probability of transitioning to a third state compared to directly transitioning between them, this instance would be taken as exhibiting a similar switching pattern as to the experiment result. Then the associated differences in the transition probabilities were extracted and averaged between two pairs. Otherwise, the difference for this instance was set to zero. This step ensured that only meaningful differences in transition probabilities were considered. By repeating this procedure 1000 times, we obtained a null distribution for the discrepancy in state transition probabilities.

## Modulation of brain state activation by time-varying stimuli features

To gain more insights into the functional nature of brain dynamic states, we investigated how narrative properties would modulate the probability of a neural state being expressed in individuals. Specifically, we focused on three different stimuli properties which were assumed to reflect an increasingly 'deeper' level of information conveyed by the narrative, including the temporal changes in acoustic property, and semantic coherence at the word level and at the clause level.

To characterize the temporal variation of acoustic property, we derived the temporal envelope of each story using the Hilbert transform, which reflects the overall fluctuation of voice amplitude. The speech envelope was then convolved with the canonical hemodynamic response function (HRF) and down-sampled to 0.5 Hz (the same resolution as the fMRI acquisition). To characterize the temporal variation of semantic coherence, we first transcribed the speech to texts and retrieved the semantic representations for each word by applying a large language model BERT that was pretrained on a large-scale Chinese corpus (*Cui et al., 2021*; *Devlin et al., 2018*). The output from the last layer of the model was used as word embedding. To avoid overfitting, we further decomposed the high-dimensional embedding vectors (N = 768) with principal component analysis (PCA) and retained the first 50 PCs (*Goldstein et al., 2024*; *Goldstein et al., 2022*). Next, a vector of word-level semantic coherence was generated for each narrative by computing the cosine similarity between the embeddings of every word and the word immediately before it. After aligning the onset time of words using Praat (https://www.fon.hum.uva.nl/praat/), the semantic coherence vector was convolved with HRF and down-sampled to 0.5 Hz. Clauses were encoded by two researchers, each including 8–9 characters on average. The semantic representations for clauses were obtained by averaging the embedding vectors of words within a clause. Using the same method, a vector for clause-level semantic coherence was generated for each narrative.

We first computed Pearson's correlation between the vector of narrative properties and the vector of state expression probability at the individual level. To infer significance, the group mean of correlation values across participants was compared to a null distribution generated by 5000 permutations. For each iteration, the time courses of brain state expression were randomly circular-shuffled, and the correlation between the narrative property vector and the shuffled time course of state expression

was re-calculated and averaged over participants to create a random value. An empirical *p*-value was determined by the proportion of values from the 5000 iterations that were larger than the original group-mean value. FDR correction was used to account for multiple comparisons.

## Correlation of latent state dynamics with behavior

To assess the importance of the timing of brain latent states to behavior, we examined whether the alignment of participant's brain state fluctuations with that of the best performer could predict their narrative comprehension scores. The best performer was the one (or those) who scored the highest in the narrative recall task within the subgroup of participants exposed to the same narrative. For each narrative, if there were more than one best performer, we first assessed the alignment between a participant with each of them, and then got the average. For the non-best-performers (N = 56), their brain alignment with the best performer(s) was measured by Pearson's correlation using the time course of state expression probability. After that, we computed a Pearson's correlation between inter-brain alignments and participants' comprehension scores.

In addition to the inter-brain alignments with the best performer, we also examined the alignment of each participant's brain state fluctuations with the group mean, which is supposed to be representative of the systematic response to the stimulus (*Nastase and Gazzola, 2019*). The inter-brain alignment was assessed by iteratively leaving out one participant and calculating the Pearson correlation between the time course of state expression probabilities for the left-out participant and the average time course for the remaining participants. All participants were engaged in the same narrative during this process.

As a comparison, we also investigated whether the overall engagement of a brain state was associated with task performance. For this purpose, we examined the correlation between participants' FO and dwell time in each state and their comprehension scores.

## Compare brain states across conditions

Finally, we investigated whether the temporal organization of brain dynamics observed during narrative comprehension was mainly driven by language processing, or instead an intrinsic organization of the dynamic brain. For this purpose, we analyzed the fMRI data from the same group of participants at rest and when listening to an unintelligible narrative told in a foreign language (Mongolian). The scanning parameters as well as the scanning length were identical to the main experiment. The HMM with K = 3 was conducted separately for the two conditions, and then the resulting three brain states were mapped to the corresponding states from the narrative comprehension condition by maximizing the similarity in state activity patterns.

## Establishing state correspondence between analyses

State correspondence between the two datasets and across different conditions was established based on spatial overlap. To assess this overlap, we first extracted each state's activity values in the nine networks derived from the HMM in the main analyses. Then, for each state identified in other analyses (treated as a candidate state), we calculated its correlation with the three states from the main analyses (treated as the predefined states) in terms of network activity patterns. The predefined state that a candidate state most closely resembled was designated to be its matching state. For instance, if a candidate state showed the strongest correlation with State #1, it was labeled as State #1 accordingly. In cases where two candidate states showed the highest similarity to the same predefined state (e.g. State #1), the candidate state with the stronger correlation was labeled as State #1, and the other was assigned to the predefined state with the next highest correlation. A prime symbol (e.g. State #2') was used to indicate cases where such confusion occurred.

Given that the spatial layout of the 9-network atlas used in the main analysis does not align well with the Yeo-7 network atlas, we established state correspondence between the two parcellation schemes using evidence of temporal overlap. Specifically, we calculated Pearson's correlation between each state derived from the Yeo-7 network scheme (treated as a 'candidate state') and the three predefined states from the main analysis, based on time courses of state expression probabilities. For each state, the time courses from all 64 participants were concatenated, resulting in 19,200 (300 × 64) time points. The predefined state most similar to each candidate state was designated as its corresponding state.

To assess the statistical significance of the spatial (and temporal) overlap, we utilized the HMM results from 1000 permutations, where participants' BOLD time courses were circularly shifted and HMMs were conducted for each permutation. Applying the same state-correspondence strategy described above, we generated a null distribution representing the matches between candidate states and states in the surrogate data.

## Acknowledgements

This work was supported by grants from the National Natural Science Foundation of China (NSFC: 32471100), Guangdong Basic and Applied Basic Research Foundation (2024A1515030046). No conflict of interest is declared. We greatly appreciate the two anonymous reviewers for their insightful comments and constructive suggestions.

## Additional information

### Funding

| Funder | Grant reference number | Author |
|---|---|---|
| Guangdong Basic and Applied Basic Research Foundation | 2024A1515030046 | Lanfang Liu |
| National Natural Science Foundation of China | 32471100 | Guosheng Ding |

The funders had no role in study design, data collection and interpretation, or the decision to submit the work for publication.

### Author contributions

Lanfang Liu, Conceptualization, Data curation, Formal analysis, Investigation, Methodology, Writing – original draft, Writing – review and editing; Jiahao Jiang, Formal analysis, Methodology, Writing – review and editing; Hehui Li, Conceptualization, Writing – review and editing; Guosheng Ding, Conceptualization, Supervision, Funding acquisition, Writing – review and editing

### Author ORCIDs

Lanfang Liu ⑩ https://orcid.org/0000-0002-1448-9009
Jiahao Jiang ⑩ https://orcid.org/0000-0001-6335-391X
Hehui Li ⑩ http://orcid.org/0000-0002-7315-3396
Guosheng Ding ⑩ http://orcid.org/0000-0002-0065-6398

### Ethics

This research was approved by the Reviewer Board of Beijing Normal University in China. Written informed consent was obtained from all participants before the study.(ICBIR_A_0002_009).

Reviewer #1 (Public review): https://doi.org/10.7554/eLife.99997.3.sa1
Reviewer #2 (Public review): https://doi.org/10.7554/eLife.99997.3.sa2
Author response https://doi.org/10.7554/eLife.99997.3.sa3

## Additional files

### Supplementary files

MDAR checklist

### Data availability

The raw and processed fMRI data are available on OpenNuero https://openneuro.org/datasets/ds005850/versions/1.0.0. The experimental stimuli and behavioral data are provided on Github at https://github.com/liulanfang11/Tripartite-brain-states (copy archived at *Liu, 2025*).

The following dataset was generated:

| Author(s) | Year | Dataset title | Dataset URL | Database and Identifier |
|-----------|------|---------------|-------------|------------------------|
| Liu L, Li H, Ding G | 2025 | Spoken narrative comprehension in Chinese | https://openneuro.org/datasets/ds005850/versions/1.0.0 | OpenNeuro, 10.18112/openneuro.ds005850.v1.0.0 |

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
