## [Editor Report · eLife Assessment]

This study provides **important** insights into the brain activity and connectivity underlying speech comprehension, revealing three brain states. The authors present **compelling** evidence by leveraging hidden Markov modeling of fMRI data to link brain state dynamics to comprehension scores, though the functional role of these states remains under-explored. These findings advance our understanding of how brain state transitions in narrative comprehension relate to stimulus-specific features.

---

## [Referee Report · Reviewer #1 (Public review)]

Summary:

Liu and colleagues applied the hidden Markov model on fMRI to show three brain states underlying speech comprehension. Many interesting findings were presented: brain state dynamics were related to various speech and semantic properties, timely expression of brain states (rather than their occurrence probabilities) was correlated with better comprehension, and the estimated brain states were specific to speech comprehension but not at rest or when listening to non-comprehensible speech.

Strengths:

Recently, the HMM has been applied to many fMRI studies, including movie watching and rest. The authors cleverly used the HMM to test the external/linguistic/internal processing theory that was suggested in comprehension literature. I appreciated the way the authors theoretically grounded their hypotheses and reviewed relevant papers that used the HMM on other naturalistic datasets. The manuscript was well written, the analyses were sound, and the results had clear implications.

---

## [Referee Report · Reviewer #2 (Public review)]

Liu et al. applied hidden Markov models (HMM) to fMRI data from 64 participants listening to audio stories. The authors identified three brain states, characterized by specific patterns of activity and connectivity, that the brain transitions between during story listening. Drawing on a theoretical framework proposed by Berwick et al. (TICS 2023), the authors interpret these states as corresponding to external sensory-motor processing (State 1), lexical processing (State 2), and internal mental representations (State 3). States 1 and 3 were more likely to transition to State 2 than between one another, suggesting that State 2 acts as a transition hub between states. Participants whose brain state trajectories closely matched those of an individual with high comprehension scores tended to have higher comprehension scores themselves, suggesting that optimal transitions between brain states facilitated narrative comprehension.

Overall, the conclusions of the paper are well-supported by the data. Several recent studies (e.g., Song, Shim, and Rosenberg, eLife, 2023) have found that the brain transitions between a small number of states; however, the functional role of these states remains under-explored. An important contribution of this paper is that it relates the expression of brain states to specific features of the stimulus in a manner that is consistent with theoretical predictions.

The correlation between narrative features and brain state expression was reliable, but relatively low (~0.03). As discussed in the manuscript, this could be due to measurement noise, as well as narrative features accounting for a small proportion of cognitive processes underlying the brain states.

A strength of the paper is that the authors repeated the HMM analyses across different tasks (Figure 5) and an independent dataset (Figure S3) and found that the data was consistently best fit by 3 brain states. Across tasks, however, the spatial regions associated with each state varied. For example, state 2 during narrative comprehension was similar to both states 2 and 3 during rest (Fig. 5A), suggesting that the organization of the three states was task dependent.

The three states identified in the manuscript correspond rather well to areas with short, medium, and long temporal timescales (see Hasson, Chen & Honey, TiCs, 2015). Given the relationship with behavior, where State 1 responds to acoustic properties, State 2 responds to word-level properties, and State 3 responds to clause-level properties, a "single-process" account where the states differ in terms of the temporal window for which one needs to integrate information over may offer a more parsimonious account than a multi-process account where the states correspond to distinct processes. This possibility is mentioned briefly in the introduction, but not developed further.

---

## [Author Response]

The following is the authors’ response to the original reviews.

**Public Reviews:**

**Reviewer #1 (Public review):**
Summary:Liu and colleagues applied the hidden Markov model on fMRI to show three brain states underlying speech comprehension. Many interesting findings were presented: brain state dynamics were related to various speech and semantic properties, timely expression of brain states (rather than their occurrence probabilities) was correlated with better comprehension, and the estimated brain states were specific to speech comprehension but not at rest or when listening to non-comprehensible speech.Strengths:Recently, the HMM has been applied to many fMRI studies, including movie watching and rest. The authors cleverly used the HMM to test the external/linguistic/internal processing theory that was suggested in comprehension literature. I appreciated the way the authors theoretically grounded their hypotheses and reviewed relevant papers that used the HMM on other naturalistic datasets. The manuscript was well written, the analyses were sound, and the results had clear implications.Weaknesses:Further details are needed for the experimental procedure, adjustments needed for statistics/analyses, and the interpretation/rationale is needed for the results.

For the Experimental Procedure, we will provide a more detailed description about stimuli, and the comprehension test, and upload the audio files and corresponding transcriptions as the supplementary dataset.

For statistics/analyses, we have reproduced the states' spatial maps using unnormalized activity pattern. For the resting state, we observed a state resembling the baseline state described in Song, Shim, & Rosenberg (2023). However, for the speech comprehension task, all three states were characterized by network activities varying largely from zero. In addition, we have re-generated the null distribution for behaviorbrain state correlations using circular shift. The results are largely consistent with the previous findings. We have also made some other adjustment to the analyses or add some new analyses as recommended by the reviewer. We will revise the manuscript to incorporate these changes.

For the interpretation/rationale: We will add a more detailed interpretation for the association between state occurrence and semantic coherence. Briefly speaking, higher semantic coherence may allow for the brain to better accumulate information over time.

State #2 seems to be involved in the integration of information at shorter timescales (hundreds of milliseconds) while State #3 seems to be involved in the longer timescales (seconds).

We greatly appreciate the reviewer for the insightful comments and constructive suggestions.

**Reviewer #2 (Public review):**
Liu et al. applied hidden Markov models (HMM) to fMRI data from 64 participants listening to audio stories. The authors identified three brain states, characterized by specific patterns of activity and connectivity, that the brain transitions between during story listening. Drawing on a theoretical framework proposed by Berwick et al. (TICS 2023), the authors interpret these states as corresponding to external sensory-motor processing (State 1), lexical processing (State 2), and internal mental representations (State 3). States 1 and 3 were more likely to transition to State 2 than between one another, suggesting that State 2 acts as a transition hub between states. Participants whose brain state trajectories closely matched those of an individual with high comprehension scores tended to have higher comprehension scores themselves, suggesting that optimal transitions between brain states facilitated narrative comprehension.Overall, the conclusions of the paper are well-supported by the data. Several recent studies (e.g., Song, Shim, and Rosenberg, eLife, 2023) have found that the brain transitions between a small number of states; however, the functional role of these states remains under-explored. An important contribution of this paper is that it relates the expression of brain states to specific features of the stimulus in a manner that is consistent with theoretical predictions.(1) It is worth noting, however, that the correlation between narrative features and brain state expression (as shown in Figure 3) is relatively low (~0.03). Additionally, it was unclear if the temporal correlation of the brain state expression was considered when generating the null distribution. It would be helpful to clarify whether the brain state expression time courses were circularly shifted when generating the null.

In the revision, we generated the null distribution by circularly shifting the state time courses. The results remain consistent with our previous findings: *p* = 0.002 for the speech envelope, *p* = 0.007 for word-level coherence, and *p* = 0.001 for clause-level coherence.

We note that in other studies which examined the relationship between brain activity and word embedding features, the group-mean correlation values are similarly low but statistically significant and theoretically meaningful (e.g., Fernandino et al., 2022; Oota et al., 2022). We think these relatively low correlations are primarily due to the high level of noise inherent in neural data. Brain activity fluctuations are shaped by a variety of factors, including task-related cognitive processing, internal thoughts, physiological states, as well as arousal and vigilance. Additionally, the narrative features we measured may account for only a small portion of the cognitive processes occurring during the task. As a result, the variance in narrative features can only explain a limited portion of the overall variance in brain activity fluctuations.

We will replace Figure 3 and the related supplementary figures with new ones, in which the null distribution is generated via circular shift. Furthermore, we will expand our discussion to address why the observed brain-stimuli correlations are relatively small, despite their statistical significance.

(2) A strength of the paper is that the authors repeated the HMM analyses across different tasks (Figure 5) and an independent dataset (Figure S3) and found that the data was consistently best fit by 3 brain states. However, it was not entirely clear to me how well the 3 states identified in these other analyses matched the brain states reported in the main analyses. In particular, the confusion matrices shown in Figure 5 and Figure S3 suggests that that states were confusable across studies (State 2 vs. State 3 in Fig. 5A and S3A, State 1 vs. State 2 in Figure 5B). I don't think this takes away from the main results, but it does call into question the generalizability of the brain states across tasks and populations.

We identified matching states across analyses based on similarity in the activity patterns of the nine networks. For each candidate state identified in other analyses, we calculate the correlation between its network activity pattern and the three predefined states from the main analysis, and set the one it most closely resembled to be its matching state. For instance, if a candidate state showed the highest correlation with State #1, it was labelled State #1 accordingly.

Each column in the confusion matrix depicts the similarity of each candidate state with the three predefined states. In Figure S3 (analysis for the replication dataset), the highest similarity occurred along the diagonal of the confusion matrix. This means that each of the three candidate states was best matched to State #1, State #2, and State #3, respectively, maintaining a one-to-one correspondence between the states from two analyses.

For the comparison of speech comprehension task with the resting and the incomprehensible speech condition, there was some degree of overlap or "confusion."

In Figure 5A, there were two candidate states showing the highest similarity to State #2. In this case, we labelled the candidate state with the strongest similarity as State #2, while the other candidate state is assigned as State #3 based on the ranking of similarity. This strategy was also applied to naming of states for the incomprehensible condition. The observed confusion supports the idea that the tripartite-state space is not an intrinsic, task-free property. To make the labeling clearer in the presentation of results, we will use a prime symbol (e.g., State #3') to indicate cases where such confusion occurred, helping to distinguish these ambiguous matches.

(3) The three states identified in the manuscript correspond rather well to areas with short, medium, and long temporal timescales (see Hasson, Chen & Honey, TiCs, 2015).Given the relationship with behavior, where State 1 responds to acoustic properties, State 2 responds to word-level properties, and State 3 responds to clause-level properties, the authors may want to consider a "single-process" account where the states differ in terms of the temporal window for which one needs to integrate information over, rather than a multi-process account where the states correspond to distinct processes.

The temporal window hypothesis provides a more fitting explanation for our results. Based on the spatial maps and their modulation by speech features, States #1, #2, and #3 seem to correspond to short, medium, and long processing timescales, respectively. We will update the discussion to reflect this interpretation.

We sincerely appreciate the constructive suggestions from the two anonymous reviewers, which have been highly valuable in improving the quality of the manuscript.

**Recommendations for the authors:**
Reviewer #1 (Recommendations for the authors):(1) The "Participants and experimental procedure" section deserves more details. I've checked Liu et al. (2020), and the dataset contained 43 participants aged 20-75 years, whereas this study contained data from 64 young adults and 30 old adult samples. The previous dataset seems to have two stories, whereas this study seems to have three. Please be specific, given that the dataset does not seem the same. Could the authors also include more descriptions of what the auditory stories were? For example, what were the contents, and how were they recorded?

The citation is partially incorrect. The dataset of young adults is shared with our work published in (2022). The 64 participants listened to one of three stories told by a female college student in Mandarin, recounting her real-life experience of hiking, a graduate admission interview, and her first time taking a flight, respectively. The sample of older adults is from our work published in (2020), which includes 30 older adults and additionally 13 young adults. The stimuli in this case were two stories told by an older woman in a Chinese dialect, describing her experience in Thailand and riding a warship, respectively. Since we aim to explore whether the main results can be replicated on a different age group, we excluded the 13 young adults from the analysis.

All the stories were recorded during fMRI scanning using a noise-canceling microphone (FOMRI-III; Optoacoustics Ltd, Or-Yehuda, Israel) positioned above the speaker’s mouth. The audio recordings were subsequently processed offline with Adobe Audition 3.0 (Adobe Systems Inc, USA) to further eliminate MRI scanner noise.

In the revised manuscript, we have updated the citation, and provided a more detailed description of the stimuli in the supplementary material. We have also uploaded the audio files along with their corresponding transcriptions to GitHub.

(2) I am curious about individual differences in comprehension scores. Did participants have less comprehension of the audio-narrated story because the story was a hard-tocomprehend narrative or because the audio quality was low? Could the authors share examples of comprehension tests?

We believe two factors contribute to the individual differences in comprehension scores. First, the audio quality is indeed moderately lower than in dailylife story-listening conditions. This is because those stories were recorded and played during fMRI scanning. Although a noise-canceling equipment was used, there were still some noises accompanying the speech, which may have made speech perception and comprehension more difficult than usual.

Second, the comprehension test measured how much information about the story (including both main themes and details) participants could recall. Specifically, participants were asked to retell the stories in detail immediately after the scanning session. Following this free recall, the experimenters posed a few additional questions drawn from a pre-prepared list, targeting information not mentioned in their recall. If participants experienced lapses of attention or did not store the incoming information into memory promptly, they might fail to recall the relevant content. In several studies, such a task has been called a narrative recall test. However, memory plays a crucial role in real-time speech comprehension, while comprehension affects the depth of processing during memory encoding, thereby influencing subsequent recall performance. To align with prior work (e.g., Stephens et al., 2010) and our previous publications, we chose to referred to this task as narrative comprehension.

In the revised manuscript, we have provided a detailed description about the comprehension test (Line 907-933) and share the examples on GitHub.

(3) Regarding Figure 3, what does it mean for a state occurrence to follow semantic coherence? Is there a theoretical reason why semantic coherence was measured and related to brain state dynamics? A related empirical question is: is it more likely for the brain states to transition from one state to another when nearby time points share low semantic similarity compared to chance?

We analyzed semantic coherence and sound envelope as they capture different layers of linguistic and acoustic structure that unfold over varying temporal scales. Changes in the sound envelope typically occur on the order of milliseconds to a few hundred milliseconds, changes in word-level semantic coherence span approximately 0.24 ± 0.15 seconds, and changes in clause-level semantic coherence extend to 3.2 ± 1.7 seconds. Previous theory and empirical studies suggest that the timescales of information accumulation vary hierarchically, progressing from early sensory areas to higher-order areas (Hasson et al., 2015; Lerner et al., 2011). Based on this work, we anticipate that the three brain states, which are respectively associated with the auditory and sensory motor network, the language network and the DMN, would be selectively modulated by these speech properties corresponding to distinct timescales.

Accordingly, when a state occurrence aligns with (clause-level) semantic coherence, it suggests that this state is engaged in processing information accumulated at the clause level (i.e., its semantic relationship). Higher coherence facilitates better accumulation, making it more likely for the associated brain state to be activated.

We analyzed the relationship between state transition probability and semantic coherence, but did not find significant results. Here, the transition probability was calculated as Gamma(t) – Gamma(t-1), where Gamma refers to the state occurrence probability. The lack of significant findings may be because brain state transitions are driven primarily by more slowly changing factors. Indeed, we found the average dwell time of the three states ranges from 9.66 to 15.29s, which is a much slower temporal dynamics compared to the relatively rapid shifts in acoustic/semantic properties.

In the revised version, we have updated the *Introduction* to clarify the rational for selecting the three speech properties and to explore their relationship with brain dynamics (Line 111-118)

(4) When running the HMM, the authors iterated K of 2 to 10 and K = 4, 10, and 12. However, the input features of the model consist of only 9 functional networks. Given that the HMM is designed to find low-dimensional latent state sequences, the choice of the number of latent states being higher than the number of input features sounds odd to me - to my speculation, it is bound to generate almost the exact same states as 9 networks and/or duplicates of the same state. I suggest limiting the K iterations from 2 to 8. For replication with Yeo et al.'s 7 networks, K iteration should also be limited to K of less than 7, or optionally, Yeo's 7 network scheme could be replaced with a 17network scheme.

We understand your concern. However, the determination of the number (K) of hidden states is not directly related to the number of features (in this case, the number of networks), but rather depends on the complexity of the time series and the number of underlying patterns. Given that each state corresponds to a distinct combination of the features, even a small number of features can be used to model a system with complex temporal behaviors and multiple states. For instance, for a system with n features, assuming each is a binary variable (0 or 1), there are maximally 2^n^ possible underlying states.

In our study, we recorded brain activity over 300 time points and used the 9 networks as features. At different time points, the brain can exhibit distinct spatial configurations, reflected in the relative activity levels of the nine networks and their interactions. To accurately capture the temporal dynamics of brain activity, it is essential to explore models that allow for more states than the number of features. We note that in other HMM studies, researchers have also explored states more than the number of networks to find the best number of hidden states (e.g., Ahrends et al., 2022; Stevner et al., 2019).

Furthermore, Ahrends et al. (2022) suggested that “Based on the HCP-dataset, we estimate as a rule of thumb that the ratio of observations to free parameters per state should not be inferior to 200”, where free parameters per state is [𝐾 ∗(𝐾 −1)+ (𝐾 −1)+𝐾 ∗𝑁 ∗(𝑁 +1)/2]/𝐾. According to this, there should be above 10, 980 observations when the number of states (K) is 10 (the maximal number in our study) and the number of networks (N) is 9. In our group-level HMM model, there were 64 (valid runs) * 300 (TR) = 19200 observations for young adults, and 50 (valid runs) * 210 (TR) = 10500 observations for older adults. Aside from the older adults' data being slightly insufficient (4.37% less than the suggestion), all other hyperparameter combinations in this study meet the recommended number of observations.

(5) In Figure 2, the authors write that the states' spatial maps were normalized for visualization purposes. Could the authors also show visualization of brain states that are not normalized? The reason why I ask is, for example, in Song, Shim, & Rosenberg (2023), the base state was observed which had activity levels all close to the mean (which is 0 because the BOLD activity was normalized). If the activity patterns of this brain state were to be normalized after state estimation, the base state would have looked drastically different than what is reported.

We derived the spatial maps of the states using unnormalized activity patterns, with the BOLD signals Z-score normalized to a mean of zero. Under the speech comprehension task, the three states exhibited relatively large fluctuations in network activity levels. The activity ranges were as follows: [-0.71 to 0.51] for State #1, [-0.26 to 0.30] for State #2, and [-0.82 to 0.40] for State #3. For the resting state, we observed a state resembling the baseline state as described in Song, Shim, & Rosenberg (2023), with activity values ranging from -0.133 to 0.09.

In the revision, we have replaced the states' spatial maps with versions showing unnormalized activity patterns.

(6) In line 297, the authors speculate that "This may be because there is too much heterogeneity among the older adults". To support this speculation, the authors can calculate the overall ISC of brain state dynamics among older adults and compare it to the ISC estimated from younger adults.

We analyzed the overall ISC of brain state dynamics, and found the ISC was indeed significantly lower among the older adults than that among the younger adults. We have revised this statement as follows:

These factors can diminish the inter-subject correlation of brain state dynamics— indeed, ISCs among older adults were significantly lower than those among younger adults (Figure S5)—and reduce ISC's sensitivity to individual differences in task performance (Line 321-326).

Other comments:(7) In Figure 4, the authors showed a significant positive correlation between head movement ISC with the best performer and comprehension scores. Does the average head movement of all individuals negatively correlate with comprehension scores, given that the authors argue that "greater task engagement is accompanied by decreased movement"?

We examined the relationship between participants' average head movement across the comprehension task and their comprehension scores. There was no significant correlation (r = 0.041, p = 0.74). In the literature (e.g. ,Ballenghein et al., 2019) , the relationship between task engagement and head movement was also assessed at the moment-by-moment level, rather than by using time-averaged data.

Real-time head movements reflect fluctuations in task engagement and cognitive state. In contrast, mean head movement, as a static measure, fails to capture these changes, and thus is not effective in predicting task performance.

(8) The authors write the older adults sample, the "independent dataset". Technically, however, this dataset cannot be independent because they were collected at the same time by the same research group. I would advise replacing the word independent to something like second dataset or replication dataset.

We have replaced the phrase “independent dataset” with “replication dataset”.

(9) Pertaining to a paragraph starting in line 586: For non-parametric permutation tests, the authors note that the time courses of brain state expression were "randomly shuffled". How was this random shuffling done: was this circular-shifted randomly, or were the values within the time course literally shuffled? The latter approach, literal shuffling of the values, does not make a fair null distribution because it does not retain temporal regularities (autocorrelation) that are intrinsic to the fMRI signals. Thus, I suggest replacing all non-parametric permutation tests with random circular shifting of the time series (np. roll in python).

In the original manuscript, the time course was literally shuffled. In the revised version, we circular-shifted the time course randomly (circshift.m in Matlab) to generate the null distribution. The results remain consistent with our previous findings: p = 0.002 for the speech envelope, p = 0.007 for word-level coherence, and p = 0.001 for clause-level coherence (Line 230-235).

(10) The p value calculation should be p = (1+#(chance>=observed))/(1+#iterations) for one-tailed test and p = (1+#(abs(chance)>=abs(observed)))/(1+#iterations) for twotailed test. Thus, if 5,000 iterations were run and none of the chances were higher than the actual observation, the p-value is p = 1/5001, which is the minimal value it can achieve.

Have corrected.

(11) State 3 in Figure S2 does not resemble State 3 of the main result. Could the authors explain why they corresponded State 3 of the Yeo-7 scheme to State 3 of the nineparcellation scheme, perhaps using evidence of spatial overlap?

The correspondence of states between the two schemes was established using evidence of state expression time course.

To assess temporal overlap, we calculated Pearson’s correlation between each candidate state obtained by the Yeo-7 scheme and the three predefined states obtained by the nine-network parcellation scheme in terms of state expression probabilities. The time courses of the 64 participants were concatenated, resulting in 19200 (300*64) time points for each state. The one that the candidate state most closely resembled was set to be its corresponding state. For instance, if a candidate state showed the highest correlation with State #1, it was labelled State #1 accordingly. As demonstrated in the confusion matrix, each of the three candidate states was best matched to State #1, State #2, and State #3, respectively, maintaining a one-to-one correspondence between the states from the two schemes.

We also assessed the spatial overlap between the two schemes. First, a state activity value was assigned to each voxel across the whole brain (including a total of 34,892 voxels covered by both parcellation schemes). This is done for each brain state. Next, we calculated Spearman’s correlation between each candidate state obtained by the Yeo-7 scheme and the three predefined states obtained by the nine-network scheme in terms of whole-brain activities. The pattern of spatial overlap is consistent with the pattern of temporal overlap, such that each of the three candidate states was best matched to State #1, State #2, and State #3, respectively.

We noted that the networks between the two schemes are not well aligned in their spatial location, especially for the DMN (as shown below). This may lead to the low spatial overlap of State #3, which is dominated by DMN activity. Consequently, establishing state correspondence based on temporal information is more appropriate in this context. We therefore only reported the results of temporal overlap in the manuscript.

We have added a paragraph in the main text for “Establishing state correspondence between analyses” (Line 672-699). We have also updated the associated figures (Fig.S2, Fig.S3 and Fig.5)

**Author response image 2. sa3fig2:** 

(12) Line 839: gamma parameter, on a step size of?(16) Figure 3. Please add a legend in the "Sound envelope" graph what green and blue lines indicate. The authors write Coh(t) and Coh(t, t+1) at the top and Coh(t) and Coh(t+1) at the bottom. Please be consistent with the labeling. Shouldn't they be Coh(t-1, t) and Coh(t, t+1) to be exact for both?

Have corrected.

(17) In line 226, is this one-sample t-test compared to zero? If so, please write it inside the parentheses. In line 227, the authors write "slightly weaker"; however, since this is not statistically warranted, I suggest removing the word "slightly weaker" and just noting significance in both States 1 and 2.

Have corrected.

(18) In line 288, please fix "we also whether".

Have corrected.

(19) In Figure 2C, what do pink lines in the transition matrix indicate? Are they colored just to show authors' interests, or do they indicate statistical significance? Please write it in the figure legend.

Yes, the pink lines indicate a meaningful trend, showing that the between-state transition probabilities are significantly higher than those in permutation.

We have added this information to the figure legend.

**Reviewer #2 (Recommendations for the authors):**
(1) It is unclear how the correspondence between states across different conditions and datasets was computed. Given the spatial autocorrelation of brain maps, I recommend reporting the Dice coefficient along with a spin-test permutation to test for statistical significance.

The state correspondence between different conditions and between the two datasets are established using evidence of spatial overlap. The spatial overlap between states was quantified by Pearson’s correlation using the activity values (derived from HMM) of the nine networks. For each candidate state identified in other analyses (for the Rest, MG and older-adult datasets), we calculate the correlation between its network activity pattern and the three predefined states from the main analysis (for the young-adults dataset), and set the one it most closely resembled to be its matching state. For instance, if a candidate state showed the highest correlation with State #1, it was labelled State #1 accordingly.

For the comparison between the young and older adults’ datasets (as shown below), the largest spatial overlap occurred along the diagonal of the confusion matrix, with high correlation values. This means that each of the three candidate states was best matched to State #1, State #2, and State #3, respectively, maintaining a one-to-one correspondence between the states from the two datasets. As the HMM is modelled at the level of networks which lack accurate coordinates, we did not apply the spin-test to assess the statistical significance of overlap. Instead, we extracted the state activity patterns from the 1000 permutations (wherein the original BOLD time courses were circularly shifted and an HMM was conducted) for the older-adults dataset. Applying the similar state-correspondence strategy, we generated a null distribution of spatial overlap. The real overlap of the three states was greater than and 97.97%, 95.34% and 92.39% instances from the permutation (as shown below).

**Author response image 3. sa3fig3:** 

For the comparison of main task with the resting and the incomprehensible speech condition, there was some degree of confusion: there were two candidate states showing the highest similarity to State #2. In this case, we labeled the most similar candidate as State #2. The other candidate was then assigned to the predefined state with which it had the second-highest correlation. We used a prime symbol (e.g., State #3') to denote cases where such confusion occurred. These findings support our conclusion that the tripartite-organization of brain states is not a task-free, intrinsic property.

When establishing the correspondence between the Yeo-7 network and the ninenetwork parcellation schemes, we primarily relied on evidence from temporal overlap measures, as a clear network-level alignment between the two parcellation schemes is lacking. Temporal overlap was quantified by calculating the correlation of state occurrence probabilities between the two schemes. To achieve this, we concatenated the time courses of 64 participants, resulting in a time series consisting of 19,200 time points (300 time points per participant) for each state. Each of the three candidate states from the Yeo-7 network scheme was best matched to State #1, State #2, and State #3 from the main analyses, respectively. To determine the statistical significance of the temporal overlap, we circular shifted each participant’s time course of state expression obtained from the Yeo-7network scheme for 1000 times. Applying the same strategy to find the matching states, we generated a null distribution of overlap. The real overlap was much higher than the instances from permutation.

**Author response image 4. sa3fig4:** 

In the revision, we have provided detailed description for how the state correspondence is established and reported the statistical significance of those correspondence (Line 671-699). The associated figures have also been updated (Fig.5, Fig. S2 and Fig.S3).

(2) Please clarify if circle-shifting was applied to the state expression time course when generating the null distribution for behavior-brain state correlations reported in Figure (3). This seems important to control for the temporal autocorrelation in the time courses.

We have updated the results by using circle-shifting to generated the null distribution. The results are largely consistent with the previous on without circular shifting (Line 230-242).

(3) Figure 3: What does the green shaded area around the sound envelope represent? In the caption, specify whether the red line in the null distributions indicates the mean or median R between brain state expression and narrative features. It would also be beneficial to report this value in the main text.

The green shaded area indicated the original amplitude of speech signal, while blue line indicates the smoothed, low-frequency contour of amplitude changes over time (i.e., speech envelope). We have updated the figure and explained this in the figure caption.

The red line in the null distributions indicates the R between brain state expression and narrative features for the real data. and reported the mean R of the permutation in the main text.

(4) The manuscript is missing a data availability statement (https://elifesciences.org/inside-elife/51839f0a/for-authors-updates-to-elife-s-datasharing-policies).

We have added a statement of data availability in the revision, as follows:

“The raw and processed fMRI data are available on OpenNeuro: https://openneuro.org/datasets/ds005623. The experimental stimuli, behavioral data and main scripts used in the analyses are provided on Github. ”

(5) There is a typo in line 102 ("perceptual alalyses").

Have corrected.

We sincerely thank the two reviewers for their constructive feedback, thorough review, and the time they dedicated to improving our work.

Reference:

Ahrends, C., Stevner, A., Pervaiz, U., Kringelbach, M. L., Vuust, P., Woolrich, M. W., & Vidaurre, D. (2022). Data and model considerations for estimating timevarying functional connectivity in fMRI. Neuroimage, 252, 119026.

Ballenghein, U., Megalakaki, O., & Baccino, T. (2019). Cognitive engagement in emotional text reading: concurrent recordings of eye movements and head motion. Cognition and Emotion.

Fernandino, L., Tong, J.-Q., Conant, L. L., Humphries, C. J., & Binder, J. R. (2022). Decoding the information structure underlying the neural representation of concepts. Proceedings of the national academy of sciences, 119(6), e2108091119. https://doi.org/10.1073/pnas.2108091119

Hasson, U., Chen, J., & Honey, C. J. (2015). Hierarchical process memory: memory as an integral component of information processing. Trends in Cognitive Sciences, 19(6), 304-313.

Lerner, Y., Honey, C. J., Silbert, L. J., & Hasson, U. (2011). Topographic mapping of a hierarchy of temporal receptive windows using a narrated story [Article]. Journal of Neuroscience, 31(8), 2906-2915. https://doi.org/10.1523/JNEUROSCI.3684-10.2011

Liu, L., Li, H., Ren, Z., Zhou, Q., Zhang, Y., Lu, C., Qiu, J., Chen, H., & Ding, G. (2022). The “two-brain” approach reveals the active role of task-deactivated default mode network in speech comprehension. Cerebral Cortex, 32(21), 4869-4884.

Liu, L., Zhang, Y., Zhou, Q., Garrett, D. D., Lu, C., Chen, A., Qiu, J., & Ding, G. (2020). Auditory–Articulatory Neural Alignment between Listener and Speaker during Verbal Communication. Cerebral Cortex, 30(3), 942-951. https://doi.org/10.1093/cercor/bhz138